# Digital economy and green total factor productivity in China

**Shuo Wang, Yueping Zheng** *, **Hailan Yang**

Business School, Shandong Jianzhu University, Jinan, China

* zhengyp1206@126.com

## Abstract

The development of information technology has created conducive conditions for the digital economy. The digital economy is regarded as a critical pathway for transforming traditional economic models. Green total factor productivity serves as an indicator for assessing the quality of economic development. During pivotal periods of economic transition, the digital economy and green total factor productivity have emerged as two prominent themes for achieving sustainable economic development. But the impact of digital economy on green total factor productivity is less discussed. Innovation environment refers to a confluence of conditions shaped by factors such as talent, funding, cultural atmosphere and government policies, all of which collectively support innovative activities within a region. The institutional environment encompasses the aggregate of economic, political, social, and legal rules. Currently, there is little discussion on bringing innovation environment and institutional environment into the impact of digital economy on green total factor productivity. To fill the research gap, this paper adopts the Slack based measure-Directional distance function model and Malmquist-Luenberger productivity index to measure green total factor productivity in each region based on the panel data collected from 30 provinces in China from 2004 to 2019. Generalized Method of Moments method is constructed to carry out an empirical study on the impact of digital economy on green total factor productivity. This paper constructs a panel threshold model with innovation environment and institutional environment as threshold variables. In further analysis, this paper employs panel quantile regression for the empirical analysis of the impact of the digital economy on green total factor productivity. Further analysis elucidates the evident disparities in the influence of the digital economy on green total factor productivity at various levels. The research results can provide a guide for discussing the green value of the digital economy and its role in fostering the development of a green economy.

## 1. Introduction

Currently, China has become the world's second-largest economy after decades of fast-paced economic development since the start of economic reforms and opening up. However, this high-speed growth of national economy has caused increasingly severe environmental

**Data Availability Statement:** All relevant data are within the manuscript and its Supporting Information files.

**Funding:** Initials of the authors who received each award: Shuo Wang Grant numbers awarded to each author: XNBS1642 The full name of each

funder: Shandong Province The funders had no role in study design, data collection and analysis, decision to publish, or preparation of the manuscript.

**Competing interests:** The authors have declared that no competing interests exist.

pollution and ecological degradation. To a large extent, these issues result from the long-standing model of extensive economic growth characterized by high growth rates, energy consumption, and emissions. Nevertheless, the emergence and development of digital technologies have empowered China to make a significant progress in growing its digital economy, such as big data, cloud computing, and artificial intelligence [1]. To further promote sustainable economic development, China has proposed the new development concept of "innovation, coordination, greenness, openness, and sharing," along with the "14th Five-Year Plan for the Development of the Digital Economy." To tackle the exacerbation of global environmental issues, it is significant for China to achieve a fine balance between economic performance and environmental preservation. Digital economy plays a key role in transforming traditional economic models and achieving green effects. In particular, green total factor productivity is essential for achieving green development, as it prioritize both environmental and economic benefits [2]. Digital economy and green total factor productivity represent the two key influencing factors in the sustainability of economic development. From a factor perspective, the development of digital economy relies on the use of data. With "green" and "clean" attributes, data enables the rapid mobilization of data resources, providing the requisite material and environmental production factors for the improvement of green total factor productivity [3, 4]. Digital technology is applicable to improve resource allocation efficiency, resource utilization, and cost reduction, thus enhancing green total factor productivity [5].

Despite prior studies conducted to reveal the impact of digital economy on green total factor productivity, there remains room for improvement in terms of indicator measurement and theoretical frameworks. Firstly, environmental pollution and resource consumption have not been included as part of the total factor productivity indicator, although China's total factor productivity have been measured in some studies [6, 7]. Therefore, traditional indicators have been modified and upgraded to measure China's green total factor productivity more accurately. Secondly, the "clean" attribute of digital economy plays a crucial role in driving the growth of green economy. However, the focus of previous research is placed on the improvement of economic efficiency [8], with the environmental perspective of the digital economy ignored. Therefore, this study aims to find out the relationship between digital economy and green total factor productivity. To achieve this purpose, the panel data from 2004 to 2019 for 30 provinces in China is used to measure their green total factor productivity respectively. Besides, Generalized Method of Moments model and threshold model are applied to empirically examine the effect of digital economy on green total factor productivity, along with the threshold effects of innovation environment and institutional environment.

The potential theoretical contributions of this study are detailed as follows. First, the DEA-SBM model is used to include relevant undesirable outputs in traditional total factor productivity, and the Malmquist-Luenberger productivity index is used to classify green total factor productivity by region. With new data perspectives provided, this contributes to the existing literature. Second, the existing literature on the green effects of digital economy is expanded from a macro perspective by exploring the impact of the digital economy on green total factor productivity. Finally, innovation environment and institutional environment are introduced as threshold variables to explore their influence exerted on the relationship between the digital economy and green total factor productivity. This is beneficial in deepening our understanding as to the green economic consequences of digital economy from the perspectives of institutional environment and innovation environment. The DEA-SBM model is a method used in the field of operations research and production management to assess the efficiency of decision-making units. DEA is a non-parametric linear programming technique that evaluates the efficiency of DMUs by comparing them against a frontier formed by the best-performing units. The DEA-SBM model addresses this issue by incorporating these slacks

directly into the efficiency score calculation. It provides a more comprehensive measure of efficiency by not only considering the radial distance but also the slacks in inputs and outputs.

The study is divided into seven sections. The literature review and research hypotheses are stated in Section 2. Section 3 presents the research design. Section 4 provides empirical analysis. Section 5 provides a discussion of the results. Section 6 is conclusions and policy implications. Finally, the limitations and further research stands in Section 7.

## 2. Literature review and research hypothesis

### 2.1 Literature review

As an emerging economy, the digital economy is having a significant impact on the way the economy is developed today. The digital economy comprises a series of economic activities that rely on digital knowledge and information as key production factors, with modern information networks serving as important carriers. Effective utilization of information and communication technology plays a critical role in driving efficiency improvements and optimizing economic structure. In contrast to traditional economies, the digital economy pervades every aspect of social production. While existing research on the digital economy has primarily centered on its economic consequences, more recent efforts have focused more directly on two critical aspects of digital economy research. The first critical aspect centers on a shift away from direct analysis of the digital economy's economic consequences to a more focused study of intermediate mechanisms. Examples of such mechanisms include the relationship between the digital economy and technological innovation, financial infrastructure, environmental regulations, foreign direct investment, and more [9–12]. The second crucial aspect involves a greater focus on exploring the interaction between the digital economy and other academic disciplines. For instance, [13] proposed that the digital economy can play a vital role in mitigating cyber threats and enhancing public welfare. Meanwhile, [14] investigated the impact of the digital economy on educational development and identified that it can significantly improve the quality of education. Notably, these studies have primarily focused on analyzing the direct or indirect impact of the digital economy on various production activities, with comparatively less attention devoted to exploring the impact of the digital economy in different contextual settings. To address this gap, the concept of green total factor productivity has emerged as an essential indicator for evaluating the quality of economic development. Green total factor productivity builds upon traditional total factor productivity accounting methodologies, incorporating factors such as resource consumption and pollution emissions to more fully capture unexpected outputs [15]. By integrating economic benefits with ecological resource protection, Green total factor productivity provides a more comprehensive evaluation of economic development. Research on green total factor productivity first involved the field of ecological and environmental. The current body of research on Green total factor productivity has predominantly explored its measurement and influencing factors. On the measurement of green total factor productivity, [16] started by introducing the DEA method into the calculation of total factor productivity. As the first to propose the directional distance function, [17] adopted it to calculate the Malmquist-Luenberger productivity index, which involves the emission of pollutants and other undesirable outputs in the total factor productivity index system. [18] developed the distance function in the non-radial SBM direction and proposed to calculate green total factor productivity by using the Luenberger index. This research has identified key factors that impact green total factor productivity, including economic factors such as capital investment and labor productivity [19], industrial structure [20], government regulations related to energy efficiency and pollution reduction [21], and environmental regulations aimed at reducing pollution and promoting sustainability [22]. For instance, [23] have

suggested that China's outward direct investment can promote the green economy at the provincial level. [24] have investigated the interplay between government and environmental decentralization and have concluded that the environment's effectiveness in promoting regional green growth is diminishing over time. However, these studies have only considered individual factors that affect green total factor productivity. There are few systematic evaluations of the digital economy's impact on green total factor productivity. Thus, this paper aims to investigate the impact of the digital economy on green total factor productivity and enrich the understanding of the factors that influence green total factor productivity at various levels.

## 2.2 Research hypothesis

**2.2.1 Digital economy and green total factor productivity.** According to the theory of technological progress, technological development plays a pivotal role in the improvement of productivity [25]. Under this context, digital technology contributes significantly to the development of digital economy. When popularized, it has the potential to enhance the efficiency of resource utilization, reduce production costs, alleviate environmental pressures, and ultimately improve the level of green total factor productivity. The theory of green growth attaches much importance to sustainable development and environmental preservation [26]. Green total factor productivity is considered as a critical indicator needed to evaluate environmental performance. In this regard, digital economy provides a large amount of data and information requisite for efficient environmental management and policy-making. To sum up, digital economy plays a pivotal role in shaping a sustainable future as it allows environmental challenges to be better understood and addressed, reduces the dependence on natural resources, mitigates pollution and carbon emissions, and enhance green total factor productivity.

First, from the perspective of technological progress, digital economy as a series of economic activities are based on the extensive application of information and communication technology. It improves green total factor productivity through the exercise of big data, cloud computing, block chain, and other digital technologies [27]. As proposed by [28], the advancement of information technology improves the level of total factor productivity. As digital economy grows, the barriers to the flow of data, information and technology in different regions are reduced. Meanwhile, technological progress is promoted by the spillover effect and demonstration effect. Digital technology is regarded as a combination of information, computing, communication and connection technologies [29, 30]. Digital economy exerts promoting effects on technological innovation. The development of digital economy reduces the risk of technological innovation, which is conducive to increasing the investment in advanced production technology, promoting the advancement of clean technology progress, and enhancing green total factor productivity. In the view of [3], the transformation and upgrading of the traditional secondary industry can be promoted through the effective integration of digital technology with traditional factors, which contributes to improving green total factor productivity and achieving green and high-quality economic development.

Second, from the perspective of factor allocation, traditional economic development is heavily reliant on the factor input growth model. However, China's economy is now gradually shifting from the rough development that pursues speed and quantity to the implicit development model that emphasizes green and quality. Digital economy represents a new factor of production. By incorporating environmental factors into the measurement framework, green total factor productivity can maximize economic benefits with such input of production factors as capital and energy [31]. Therefore, digital resources can be applied as green production factors to promote the development of green industry by means of digital technology. The allocation of capital and various factors is continuously optimized to flow towards high-efficiency

departments, which is effective in reducing the dependence on energy resources, and in promoting the upgrading of industrial structure towards digitization, rationalization, and green. Therefore, the combination of digital economy and traditional industrial production elements is capable to lower the excessive consumption of energy, reduce environmental undesired output, optimize the industrial structure, reduce the input of production factors, improve the utilization efficiency of traditional factors, and enhance regional green total factor productivity. Therefore, the following research hypothesis is proposed:

H1 Digital economy can promote green total factor productivity.

**2.2.2 The threshold effect of innovation environment.** Innovation environment refers to the regional environment created for the subjects involved in cooperative interaction and collective learning, including cultural environment, infrastructure, management system, policies and regulations, etc [32]. In the existing literature, the innovation environment is discussed. As argued by [33], financial support, talent support, government policies, and investment projects represent the important dimensions for evaluating the regional innovation environment. [34] analyzed the impact of the innovation environment on R&D from two perspectives: financial environment and human capital. [35] divide the innovation environment into financial environment, learning environment and cultural environment to measure the level of regional innovation. The innovation environment plays a vitally important role in improving innovation efficiency, supporting innovation activities, and enhancing green total factor productivity [36].

The Schumpeterian innovation theory emphasizes the critical role played by the innovation environment in shaping and influencing innovation in various aspects, including its types, quantity, and effects [37]. A favorable innovation environment is conducive to the development of digital technology and the upgrading of digital infrastructure, both of which are essential for growing the digital economy. Determining whether digital transformation can be effectively driven in a region, the level of innovation environment has an immediate effect on the progress of digital economy [38]. In the regions where a certain threshold is not met in terms of their innovation environment, limited resources may be allocated to other priorities, which hinders the construction of digital infrastructure and the effort made to create the green value associated with digital economy. Conversely, in the regions where the level of innovation environment is higher, distinct advantages are enjoyed in terms of digital information infrastructure construction and financial support. Due to these advantages, innovative digital technologies are applied to enhance productivity, reduced resource waste, lower production costs, and ultimately improve green total factor productivity. Based on the above analysis, the following hypotheses are put forward:

H2 Digital economy has the threshold effect of innovation environment on green total factor productivity for China.

**2.2.3 The threshold effect of institutional environment.** According to new institutional economics, the institutional environment, as an external macro factor, exerts a significant influence on the behavior and decision-making of agents. There is a close association between the degree of institutional development and the level of marketization within a region. The growth of digital economy hinges on various inputs, such as research capital, innovation activities, regional economic development, and institutional environment. When the institutional environment varies, the effect of digital economy on green total factor productivity shows nonlinearity. A more favorable institutional environment implies a market that is fairer and more

rational. In this context, the government bases the allocation of innovation resources on market size and the intensity of innovation input, rather than resorting to institutionalized special arrangements. In the regions with a better institutional environment, speculative behavior is suppressed, equal competition opportunities are provided, and green total factor productivity is further enhanced. By promoting the healthy development of market intermediaries, a favorable institutional environment enables the efficient flow of material and information, including technology, talent, knowledge, and data. To a certain extent, this is conducive to overcoming the obstacles in the development of digital economy, promoting the innovation in digital technology, optimizing the allocation of green resources, and thus enhancing green total factor productivity. However, in the regions with inadequate institutions, preferential policies are not available, and rent-seeking behavior is prevalent, which undermines regional digital technology innovation and progress. Consequently, the positive impact of digital economy on green total factor productivity diminishes. Based on the above analysis, the following hypotheses are proposed:

H3 Digital economy has the threshold effect of institutional environment on green total factor productivity for China.

## 3. Research design

### 3.1 Research model

The dynamic effect of digital economy on green total factor productivity are investigated in this paper. According to [39], we set up the the Generalized Method of Moments (Generalized Method of Moments) to test hypothesis1.

$$gtfp_{it} = \alpha_0 + \alpha_1 de_{it} + \alpha_2 control_{it} + \lambda_i + \varepsilon_{it} \tag{1}$$

Where i denotes province and t denotes year, the dependent variable $gtfp_{it}$ is the green total factor productivity, $gtfp_{it-1}$ represents the lagging term of green total factor productivity, the independent variable is $de_{it}$ that represents the digital economy, $\lambda_i$ represents the regional effect, and $\varepsilon_{it}$ represents the random term of the model, and the rest are control variables.

$$gtfp_{it} = \alpha_0 + \alpha_1 gtfp_{it-1} + \alpha_2 de_{it} + \alpha_3 control_{it} + \lambda_i + \varepsilon_{it} \tag{2}$$

To further explore the different relationships between digital economy and green total factor productivity, Hansen's method is taken as reference while innovation environment and institutional environment are treated as threshold variables in this study [40]. On this basis, a panel threshold regression model is constructed to thoroughly analyze the nonlinear impact of the digital economy on green total factor productivity. The impact of digital economy on green total factor productivity is determined by the value interval of the threshold variable digital economy. In accordance with H2 and H3, the threshold values of innovation environment (inv) and institutional environment (ins) are set, with γ representing the estimated threshold, and I(·) referring to the indicator function. If the equation in the brackets is true, then I is 1, and it is 0 otherwise. The model is expressed as follows:

$$gtfp_{it} = \alpha + \beta_1 de_{it} \times I(inv_{it} \leq \gamma) + \beta_2 de_{it} \times I(inv_{it} > \gamma) + \beta_3 X_{control} + u_i + \varepsilon_{it} \tag{3}$$

$$gtfp_{it} = \alpha + \beta_1 de_{it} \times I(ins_{it} \leq \gamma_1) + \beta_2 de_{it} \times I(\gamma_1 < ins_{it} \leq \gamma_2) + \beta_3 de_{it} \times I(ins_{it} > \gamma_2) + \beta_4 X_{control} + u_i + \varepsilon_{it} \tag{4}$$

## 3.2 Data source and sample selection

The panel data of 30 province-level provinces in China are used in this study during the period of 2004–2019(excluding Tibet Autonomous Region, Hong Kong Special Administrative Region, Macau Special Administrative Region and Taiwan Province due to missing data). The development indexes of digital economy are selected from the Chinese Research Data Services Platform which provides economic information services in China. The database can be searched on cnrds. com. Green total factor productivity data are selected from the China Statistical Yearbook and the China Statistical Yearbook on Environment. China Statistical Yearbook comprehensively reflects the statistical data of economic and social development of various provinces and regions in China. China Statistical Yearbook on Environment predominantly centers on environmental statistical data and pertinent information within China. It encompasses statistical data pertaining to environmental conservation, resource utilization, pollution control, and ecological conditions. Financial data are collected from the China Stock Market and Accounting Research Database. China Stock Market and Accounting Research Database is an important financial information database in China, associated with Guo-Tai-An firm. Primarily, the database collects, organizes, and disseminates various financial and economic data to facilitate risk assessment, market analysis, policy formulation, and decision-making for financial institutions, investors, researchers.

**3.2.1 Explained variable.** Compared with the total factor productivity, the green total factor productivity takes into account unexpected outputs such as energy consumption and environmental pollution, and can more accurately evaluate the growth efficiency of the green economy. Referring to the ideas of [18], the ML index method under the non-radial and non-angle SBM directional distance function in this study is used to measure the green total factor productivity in China.

Based on SBM-DDF model, this paper regards each province as a production decision-making unit ($DMU_j = 1,2,3,\ldots,k$). Each DMU contains input, expected output and unexpected output. X represents m inputs for each unit, y represents n expected outputs, and u represents k unexpected outputs. This paper defines the matrix as follows: $X = [x_1, x_2, \ldots, x_k] \in R^{mk}$; $Y = [y_1, y_2, \ldots, y_k] \in R^{nk}$; $U = [u_1, u_2, \ldots, u_k] \in R^{kk}$. And this paper defines the function as follows:

$$\text{Min } \rho = \frac{1 - \frac{1}{M}\sum_{m=1}^{M}\frac{S_m^x}{x_{m0}}}{1 + \frac{1}{N+H}\left(\sum_{n=1}^{N}\frac{S_{n0}^y}{y_{n0}} + \sum_{k=1}^{H}\frac{S_k^u}{u_{k0}}\right)}$$

$$\text{s.t.} \sum_{K=1}^{K} z_k x_{mk} + sx_m = x_{m0}, m = 1, 2, \ldots, M$$

$$\sum_{K=1}^{K} z_k u_{nk} - sy_n = y_{n0}, n = 1, 2, \ldots, M$$

$$\sum_{K=1}^{K} z_k u_{nk} + su_k = u_{k0}, h = 1, 2, \ldots, M$$

$$\sum_{K=1}^{K} z_k = 1, z_k, s_m^x, s_n^y, s_k^u \geq 0 \tag{5}$$

In Formula ([5]), ρ represents the frontier static efficiency of input and output. s represents the slack variable of input and output, where output contains expected output and unexpected output. $z_k$ represents the weight of each decision-making unit. Referring to the research of [18], this study decomposes the inefficiency value as follows:

$$IE = \frac{1}{M}\sum_{m=1}^{M}\frac{S_m^x}{x_{m0}}$$

$$OE = \frac{1}{N+H}\sum_{n=1}^{N}\frac{S_m^y}{y_{n0}} \tag{6}$$

After the directional distance function is obtained by solving linear programming, the SBM-DDF model considering unexpected output is combined with the ML productivity index to measure green total factor production in this paper. Referring to the research of [17], the ML index from period *t* to period *t + 1* is:

$$ML\_GTFP_t^{t+1} = \left[\frac{(1 + \rightarrow D_0^t(x^t, y^t, b^t; y^{t+1}; g^{t+1}))}{(1 + \rightarrow D_0^t(x^{t+1}, y^{t+1}, b^{t+1}; y^{t+1}; g^{t+1}))} \times \frac{(1 + \rightarrow D_0^{t+1}(x^t, y^t, b^t; y^{t+1}; g^{t+1}))}{(1 + \rightarrow D_0^{t+1}(x^{t+1}, y^{t+1}, b^{t+1}; y^{t+1}; g^{t+1}))}\right]^{1/2} \tag{7}$$

$$ML_t^{t+1} = ML\_EFFCH \times ML\_TECH \tag{8}$$

Where, the ML index represents the change rate of green total factor productivity. And the ML index can be further decomposed into the technical efficiency change index (*ML _ EFFCH*) and the technical progress change index (*ML _TECH*). When *ML _GTFP>1*, *ML _ EFFCH >1*, *ML _ TECH >1*, it indicates that the GTFP increases from period *t* to period *t + 1*. When *ML _GTFP<1*, *ML _ EFFCH <1*, *ML _ TECH <1*, it indicates that the green total factor productivity decreases from period *t* to period *t + 1*.

In this paper, the following input, expected output and unexpected output indicators are used to measure the green total factor productivity of each province by using SBM-DDF and Malmquist-Luenberger index. Input indicators: (1) Capital input: This study uses the perpetual inventory method to calculate the capital stock of each province at constant prices after estimating a base year by referring to [41]. The formula is: $K_{it} = K_{it-1}(1-\delta)+I_{it}$. Among them, $I_{it}$ is the total amount of fixed assets formed in the current year, δ is the depreciation rate (here, at 9.6%). (2) Energy input: Energy consumption is used as an indicator to measure energy input. (3) Labor input: Measured by the employees in urban units. The expected output index is characterized by the real GDP and adjusted to a constant price in 2004 according to the GDP deflator. Unexpected output indicators are measured by industrial wastewater, industrial sulfur dioxide and industrial smoke emissions.

Green total factor productivity is an indicator that measures production efficiency by considering both the resources required for production and the environmental impact of output. The primary resource inputs considered in green total factor productivity are capital, energy, and labor. The selection of capital, energy, and labor is driven by several reasons. First, these elements constitute the core components of the production process. Capital encompasses equipment, machinery, and other investments; energy stands as an indispensable resource in production activities, while labor represents human workforce. Evaluating these elements allows for a more comprehensive assessment of production efficiency and resource utilization within the production process. Second, the inclusion of energy as a factor is due to its close association with environmental impact. Assessing energy inputs aids in measuring the production's environmental footprint. Considering energy use allows for a better understanding of

whether production is sufficiently efficient, thereby reducing resource wastage and environmental pollution. Third, capital and labor are comparable resource elements. This comparability facilitates easier comparisons of production efficiency among different industries, countries, or regions. With these elements sharing common units of measurement, it aids in analysis and research. Therefore, utilizing capital, energy, and labor as constituents of green total factor productivity contributes to a comprehensive evaluation of production efficiency while integrating environmental factors, thereby better supporting sustainable and environmentally friendly economic growth.

Based on the above input and output indicators, the green total factor productivity of 30 provinces in China from 2004 to 2019 was calculated by using the MaxDEA software. The ML index reflects the change rate of green total factor productivity in the current year relative to the previous year, therefore. The green total factor productivity of the base period in 2004 is set to 1. Then we multiply this with ML index of each year in turn to obtain the green total factor productivity of each year.

**3.2.2 Explanatory variable.** Referring to the research by [17, 42], this paper uses the internet related output, related employees, the internet penetration status and mobile phone penetration status to measure the level of digital economy. The specific indicators are: the per capita telecommunication service, the proportion of computer service and software employees, the number of internet broadband access users per 100 people, the number of mobile phone users at the end of the year. In the calculation process, the topsis entropy weight method is used to calculate the digital economy index at the province level in China.

Based on the meaning of digital economy, the Internet is an important carrier for the development of digital economy. The construction of digital economy indicators at the inter-provincial level takes Internet informatization as the core of the measurement. For example, [43] published an article on constructing the digital economy index through informatization and digitization indicators. For the selection of digital economy indicators, in line with the literature [44], our indicators include both total telecommunication services per capita, the share of the number of people in information computer services, and other information technology indicators. Informationization indicators reflect the current degree of informationization in the region and reflect the basic support environment of the development of the digital economy. Our indicators also include the number of Internet broadband access users, the number of cell phone users and other Internet development indicators, which fully consider the infrastructure of the digital economy and reflect the conditions for the development of the digital economy.

**3.2.3 Threshold variables.** Referring to the research by [45], this paper uses government financial expenditure on science and technology, information infrastructure and financial development to measure the level of innovation environment. Regions with high level of innovation environment have more scale advantage in digital technology research and development as well as financial support. When the level of regional innovation environment is higher, the better the digital infrastructure is, and there is more incentive to use digital technology for secondary industry transformation and upgrading. Therefore, when the innovation environment exceeds the threshold value there will be an impact on green total factor productivity. The details are shown in Table 1.

A good institutional environment will reduce government intervention in the market, stimulate market vitality, attract investment, and stimulate green technology R&D investment and innovation. Therefore, the institutional environment can affect green total factor productivity. Referring to the research by [46]. This paper uses the market index of Fan gang to measure the institutional environment. The institutional environment constitutes an external factor that influences corporate behavior and decision-making. A favorable institutional environment

**Table 1. Innovation environment variable measurement.**

| Primary indexes | Secondary indexes | Three-level indexes |
|---|---|---|
| Innovation environment | Government financial expenditure on science and technology | The proportion of government expenditure on science and technology to total financial expenditure |
| | Information infrastructure | The per capita internet access ports×0.5+ The per capita Mobile telephone exchange capacity×0.5 |
| | Financial development | The proportion of various RMB loan balance of financial institutions to GDP at the end of the year |

implies equitable and transparent market competition. The degree of institutional environment improvement depends on the level of regional marketization. The Marketization Index of China's Provinces, proposed by Fan Gang, is a metric for assessing the level of marketization in various regions of China. The comprehensive index takes into account several factors, including government-market relationships, non-state economic development, product market development, market service scale, and legal environment. The Index serves as a tool for governmental bodies, research institutions, and economists to gain insights into the marketization status of different regions in China, facilitating the formulation of policies aimed at promoting economic efficiency and growth. The Index, proposed by Fan Gang, can be accessed on cmi.ssap.com.cn.

**3.2.4 Control variables.** We control for eight factors known to affect green total factor productivity. The following variables are controlled based on the studies of [47–49].

(1) Human capital. High-tech talents have the ability to innovate green technologies, and the improvement of human capital has an impact on green total factor productivity. This paper uses the per capita education years to measure. (2) Traffic condition. Areas with convenient transportation can strengthen the level of interconnection between regions, enhance the flow of factors between regions, and thus improve green total factor productivity. This paper uses the per capita road area of each province to measure the level of transportation condition. (3) Industry structure. Regions with a reasonable industry structure will pay more attention to the development of green industries and energy industries, thereby promoting the growth of green economy. (4) Research and development investment. Areas with more R&D investment have stronger green technology innovation capabilities, which will affect the improvement of green total factor productivity. This paper uses the R&D expenditure of each province to measure. (5) The level of economic development. Regions with a high level of economic development have advantageous production factors such as capital and technology, which are conducive to promoting the development of green economy in various regions. This paper uses the per capita GDP of each province to measure. (6) Foreign direct investment. Foreign investment in regional green innovation affects green total factor productivity. In this paper, we measure foreign direct investment as a share of GDP. (7) The level of science and technology. Faster factor mobility between more open regions is conducive to higher green total factor productivity. In this paper, total exports and imports are measured as a share of regional GDP. (8) The level of opening up. The higher the level of science and technology, the stronger the green technology innovation capacity, thus affecting the green total factor productivity. This paper measures the share of science and technology expenditures in fiscal expenditures.

We choose control variables for the following reasons: First, it is imperative to explicitly define the research inquiry concerning the relationship between the digital economy and green total factor productivity, identifying the theoretical factors associated with both the digital economy and green total factor productivity. These factors are commonly perceived as potential control variables. Consequently, our choice of control variables is grounded in the theoretical underpinnings of the research. Second, a review of the literature reveals the control

variables utilized by other researchers in exploring the nexus between the digital economy and green total factor productivity. References can be used as the basis for selecting control variables [47, 49]. Third, there may exist endogeneity issues between the digital economy and green total factor productivity. Control variables are instrumental in mitigating bias stemming from endogeneity. To address this, we have selected Control Variables (Human capital, Traffic condition, Industry structure, Research and development investment, The level of economic development, Foreign direct investment, The level of science and technology, The level of opening up) to alleviate endogeneity concerns.

These variables are not the sole determinants affecting green total factor productivity. We posit the existence of latent variables influencing green total factor productivity. However, in this study, we opted for a Generalized Method of Moments to mitigate the impact of latent variables. Generalized Method of Moments allows for the consideration of the lagged effects of green total factor productivity, enabling a more comprehensive capture of latent variables that might manifest effects across multiple time periods. By introducing the lagged term of green total factor productivity as a control variable, Generalized Method of Moments provides a more holistic assessment of the influence of latent variables.

## 4. Empirical analysis

### 4.1 Distribution of digital economy and green total factor productivity in China

The ML index reflects the change rate of green total factor productivity. This paper uses 2004 as the base period to obtain the ML index of each province in China during the study period. Table 2 shows the distribution of digital economy in China from 2004–2019. Table 2 shows the distribution of green total factor productivity in China from 2004–2019. From a regional perspective, it can be seen from Table 2 that, due to factors such as good geographical location, advanced technology and higher economic level, digital economy shows a spatially uneven feature of higher in the east and relatively lower in the middle and west during 2004–2019. The eastern region uses digital technology to develop a high value-added economy, rationally utilizes green resources and optimizes the industrial energy structure, which is conducive to green development. The development of the digital economy in the central and western regions is weak. The application of digital clean technology to traditional industries is not sufficient, and green total factor productivity needs to be improved. As shown in Table 2, the green total factor productivity in the eastern region is higher than that in the central and western regions, and shows the spatial characteristics of decreasing from the eastern coast to the northwest inland.

### 4.2 Descriptive statistics

The variables, the indicators and the results of descriptive statistics are provided in Table 3.

### 4.3 Vif test

VIF tests of the main variables are provided in Table 4, intending to preliminarily check for the presence of multicollinearity. To verify there is no multicollinearity, the VIF test was conducted in this study. All variables have VIF < 5, and the mean VIF is 1.980, indicating that multicollinearity is not a problem.

### 4.4 Regression analysis

To test the appropriateness of the Generalized Method of Moments method setting, this paper conducts regression analyses on the lagged data of digital economy and green total factor

**Table 2. Table of average values of digital economy and green total factor productivity in China's Provinces from 2004 to 2019.**

| Province | Green Total Factor Productivity | Digital Economy |
|---|---|---|
| Beijing | 1.0781 | 0.1611 |
| Tianjin | 1.0690 | 0.0181 |
| Hebei | 1.0523 | 0.0397 |
| Shanxi | 1.0509 | 0.0160 |
| Inner Mongolia | 1.0950 | 0.0216 |
| Liaoning | 1.0678 | 0.0335 |
| Jilin | 1.0506 | 0.0205 |
| Heilongjiang | 1.0152 | 0.0162 |
| Shanghai | 1.0666 | 0.0532 |
| Jiangsu | 1.0929 | 0.0756 |
| Zhejiang | 1.0477 | 0.0710 |
| Anhui | 1.0437 | 0.0195 |
| Fujian | 1.0377 | 0.0276 |
| Jiangxi | 1.0597 | 0.0141 |
| Shandong | 1.0712 | 0.0607 |
| Henan | 1.0423 | 0.0445 |
| Hubei | 1.0564 | 0.0223 |
| Hunan | 1.0572 | 0.0240 |
| Guangdong | 1.0364 | 0.1747 |
| Guangxi | 1.0294 | 0.0308 |
| Hainan | 1.0592 | 0.0226 |
| Chongqing | 1.0778 | 0.0173 |
| Sichuan | 1.0556 | 0.0408 |
| Guizhou | 1.0554 | 0.0253 |
| Yunnan | 1.0331 | 0.0250 |
| Shaanxi | 1.0573 | 0.0293 |
| Gansu | 1.0355 | 0.0135 |
| Qinghai | 1.0539 | 0.0256 |
| Ningxia | 1.0844 | 0.0210 |
| Xinjiang | 1.0508 | 0.0248 |

productivity across 30 provinces in China using ordinary least square, fixed effect, and Generalized Method of Moments method. The outcomes of the regression are presented in Table 5. The results indicate that the coefficients of L.gtfp factor productivity are significant, suggesting an impactful relationship from the previous period's green total factor productivity on the current period. The L.gtfp coefficient of ordinary least square and fixed effect are 0.1631 and 0.1146, respectively. The L.gtfp coefficient of Generalized Method of Moments method is 0.1558, which lies between the ordinary least square and fixed effect. It shows that the model fitting effect is good. In this paper, we choose Generalized Method of Moments method for estimation.

In order to reflect the fact that green total factor productivity has dynamic characteristics over time, this study uses Generalized Method of Moments method. The two tests are applied to ensure the accuracy of empirical results. First, this study examines the significance of the first-order autocorrelation AR(1) and second-order autocorrelation AR(2), which are shown in Table 6. AR(1) is significant, while AR(2) is not significant. These results indicate that the second-order sequence correlation is not significant. Over-recognition is analyzed by the

**Table 3. Descriptive statistics.**

| Variable | Indicator | Obs | Mean | Std. Dev. | Min | Max |
|---|---|---|---|---|---|---|
| gtfp | Green total factor productivity | 480 | 1.056 | 0.091 | 0.747 | 1.641 |
| de | Digital economy | 480 | 0.039 | 0.063 | 0.001 | 0.551 |
| road | Traffic condition | 480 | 13.720 | 4.649 | 4.040 | 26.200 |
| indu | Industry structure | 480 | 0.229 | 0.254 | 0.008 | 1.214 |
| edu | Human capital | 480 | 8.768 | 1.018 | 6.378 | 12.850 |
| rd | Research and development investment | 480 | 14.120 | 1.509 | 9.677 | 17.250 |
| pgdp | The level of economic development | 480 | 10.390 | 0.698 | 8.346 | 12.010 |
| for | Foreign direct investment | 480 | 0.777 | 0.879 | 0.001 | 6.420 |
| tech | The level of science and technology | 480 | 2.025 | 1.389 | 0.390 | 7.900 |
| open | The level of opening up | 480 | 0.308 | 0.375 | 0.013 | 1.722 |

Hansen test. The empirical results of the Hansen test cannot reject the null hypothesis at the 10% significance level. Therefore, instrumental variables with first-order lag are effective and there is no problem of over-identification. These results indicate that there is an obvious temporal continuity between the current green total factor productivity and the previous period. Therefore, we further examine the validity of the Generalized Method of Moments method.

Table 6 reports the regression results of the impact of digital economy on green total factor productivity. We found that there is a significant positive correlation between digital economy and green total factor productivity at the 10% level, which indicates that the digital economy promotes the green total factor productivity. Thus, the Hypothesis 1 is supported.

## 4.5 Threshold regression of digital economy on green total factor productivity

The previous analysis shows that the digital economy has a promoting effect on green total factor productivity. However, there are differences in regional factor endowment characteristics in China. The digital economy is constrained by many factors in order to play its role. Therefore, the relationship between the digital economy and green total factor productivity is nonlinear. Referring to the research by [40], we test the threshold effect of the model in this study. The model is estimated using bootstrap method sampling 300 times with innovation environment and institutional environment as threshold variables.

In this paper, single, double and triple thresholds are tested in turn and the threshold values and the statistics are derived. As shown in Tables 7 and 8, the threshold variable of the

**Table 4. Vif test.**

| Variable | vif | 1/vif |
|---|---|---|
| de | 1.270 | 0.785 |
| road | 1.330 | 0.750 |
| indu | 1.610 | 0.623 |
| edu | 2.350 | 0.426 |
| rd | 2.370 | 0.421 |
| pgdp | 3.680 | 0.272 |
| for | 1.670 | 0.600 |
| tech | 2.370 | 0.421 |
| open | 1.130 | 0.884 |
| mean vif | 1.980 | |

**Table 5. Model consistency test.**

|  | Mixed regression | Fixed effect regression | SYS-Generalized Method of Moments |
|---|---|---|---|
|  | (1) | (2) | (3) |
| L.gtfp | 0.1631** | 0.1146** | 0.1558* |
|  | (0.0496) | (0.0413) | (0.0791) |
| de | 0.2982** | 0.5143*** | 0.4570** |
|  | (0.1270) | (0.1333) | (0.1809) |
| rd | -0.0025 | 0.0001 | -0.0009 |
|  | (0.0026) | (0.0026) | (0.0033) |
| road | 0.0027** | 0.0035** | 0.0041** |
|  | (0.0011) | (0.0010) | (0.0016) |
| indu | -0.0226 | -0.0162 | -0.0160 |
|  | (0.0187) | (0.0193) | (0.0262) |
| edu | -0.0111* | -0.0139* | -0.0161* |
|  | (0.0059) | (0.0070) | (0.0091) |
| pgdp | 0.0073 | 0.0014 | 0.0022 |
|  | (0.0094) | (0.0095) | (0.0139) |
| open | -0.0115 | 0.0558 | -0.0184 |
|  | (0.0071) | (0.0344) | (0.0132) |
| tech | 0.0079 | 0.0086 | 0.0101 |
|  | (0.0060) | (0.0069) | (0.0092) |
| for | 0.0071 | 0.0068 | 0.0056 |
|  | (0.0043) | (0.0045) | (0.0054) |
| cons | 0.8831*** | 0.9386*** | 0.9373*** |
|  | (0.0892) | (0.0939) | (0.1300) |
| $R^2$ | 0.076 | 0.093 |  |
| AR(1) |  |  | 0.001 |
| AR(2) |  |  | 0.455 |
| Hansen test |  |  | 0.452 |

Note

*** p<0.01

** p<0.05

* pa<0.1. Hansen statistic to test whether the moment condition has an over-identification problem to determine whether the instrumental variable is valid. AR(1) is the residual first-order serial correlation test. AR(2) is the residual second-order serial correlation test.

innovation environment passes the single threshold test at the 10% level. The threshold value of the innovation environment is 0.0662. The confidence interval is from 0.0394 to 0.0673. The threshold variable of the institutional environment passes the double threshold test at the 5% level. The threshold values of the institutional environment are 4.510 and 4.740. The confidence intervals are [3.9550, 4.560], [4.5250, 4.7900]. As shown in Figs 1 and 2, in order to judge the distribution of threshold values and confidence intervals more clearly, this paper shows the likelihood ratio graphs with innovation environment and institutional environment as threshold variables.

In this research, the model (13) and the model (14) are tested for the threshold. And the specific parameter estimation results are shown in Table 9:

The results of the threshold model in the research design of this paper are shown in Table 9, indicating that the coefficient of the digital economy is -0.313 when the innovation environment is lower than the threshold value of 0.0662. Within the first threshold range, the digital economy exerts a significant negative effect on green total factor productivity in China. The

**Table 6. Regression analysis of the relationship among digital economy and green total factor productivity.**

|  | (4) | (5) | (6) | (7) | (8) | (9) | (10) | (11) | (12) |
|---|---|---|---|---|---|---|---|---|---|
| L.gtfp | 0.1523** | 0.1658** | 0.1525** | 0.1672** | 0.1640** | 0.1621** | 0.1512** | 0.1432* | 0.1558* |
|  | (0.0551) | (0.0542) | (0.0542) | (0.0579) | (0.0632) | (0.0632) | (0.0713) | (0.0727) | (0.0791) |
| de | 0.3016** | 0.3588** | 0.3499** | 0.3669** | 0.3724** | 0.3791** | 0.4209** | 0.4227** | 0.4570** |
|  | (0.1214) | (0.1357) | (0.1390) | (0.1391) | (0.1455) | (0.1560) | (0.1643) | (0.1646) | (0.1809) |
| road |  | 0.0035** | 0.0038** | 0.0043** | 0.0043** | 0.0040** | 0.0043** | 0.0044** | 0.0041** |
|  |  | (0.0012) | (0.0011) | (0.0015) | (0.0016) | (0.0015) | (0.0016) | (0.0015) | (0.0016) |
| edu |  |  | -0.0079* | -0.0075* | -0.0077* | -0.0080* | -0.0075* | -0.0078* | -0.0161* |
|  |  |  | (0.0041) | (0.0042) | (0.0043) | (0.0039) | (0.0040) | (0.0042) | (0.0091) |
| rd |  |  |  | -0.0029 | -0.0024 | -0.0021 | -0.0018 | -0.0019 | -0.0009 |
|  |  |  |  | (0.0022) | (0.0023) | (0.0038) | (0.0038) | (0.0038) | (0.0033) |
| indu |  |  |  |  | 0.0079 | 0.0102 | 0.0112 | 0.0052 | -0.0160 |
|  |  |  |  |  | (0.0192) | (0.0194) | (0.0194) | (0.0187) | (0.0262) |
| pgdp |  |  |  |  |  | 0.0006 | -0.0000 | 0.0022 | 0.0022 |
|  |  |  |  |  |  | (0.0120) | (0.0122) | (0.0136) | (0.0139) |
| open |  |  |  |  |  |  | -0.0169 | -0.0178 | -0.0184 |
|  |  |  |  |  |  |  | (0.0109) | (0.0128) | (0.0132) |
| for |  |  |  |  |  |  |  | 0.0048 | 0.0056 |
|  |  |  |  |  |  |  |  | (0.0051) | (0.0054) |
| tech |  |  |  |  |  |  |  |  | 0.0101 |
|  |  |  |  |  |  |  |  |  | (0.0092) |
| cons | 0.8873*** | 0.8216*** | 0.9006*** | 0.9142*** | 0.9107*** | 0.9086*** | 0.9167*** | 0.9027*** | 0.9373*** |
|  | (0.0577) | (0.0557) | (0.0657) | (0.0708) | (0.0759) | (0.1048) | (0.1076) | (0.1338) | (0.1300) |
| AR(1) | 0.001 | 0.001 | 0.001 | 0.001 | 0.001 | 0.001 | 0.001 | 0.001 | 0.001 |
| AR(2) | 0.112 | 0.228 | 0.119 | 0.220 | 0.245 | 0.227 | 0.217 | 0.251 | 0.455 |
| Hansen test | 0.372 | 0.381 | 0.371 | 0.380 | 0.382 | 0.414 | 0.417 | 0.438 | 0.452 |

Note

*** $p < 0.01$

** $p < 0a.05$

* $p < 0.1$. Hansen statistic to test whether the moment condition has an over-identification problem to determine whether the instrumental variable is valid. AR(1) is the residual first-order serial correlation test. AR(2) is the residual second-order serial correlation test.

**Table 7. Threshold effect test.**

| Threshold variable | Number of threshold | P value | F value | BS times | Critical value | | |
|---|---|---|---|---|---|---|---|
|  |  |  |  |  | 1% | 5% | 10% |
| Innovation environment | Single threshold | 0.0567 | 13.44* | 300 | 20.0357 | 13.4957 | 11.5288 |
|  | Double threshold | 0.6000 | 5.07 | 300 | 24.4796 | 16.4280 | 12.7115 |
| Institutional environment | Single threshold | 0.0667 | 11.87* | 300 | 29.3833 | 14.4796 | 9.2843 |
|  | Double threshold | 0.0300 | 17.98** | 300 | 23.0257 | 12.6555 | 10.1840 |
|  | Triple threshold | 0.6267 | 6.68 | 300 | 34.9180 | 22.2771 | 17.3663 |

Note

*** $p < 0.01$

** $p < 0.05$

* $p < 0.1$. Both p-values and critical values are obtained by repeatedly sampling 300 times using the bootstrap method.

**Table 8. Threshold effect test.**

| Threshold variable | Number of threshold | Estimated value | Confidence interval |
|---|---|---|---|
| Innovation environment | Single threshold | 0.0662 | [0.0394, 0.0673] |
| Institutional environment | Double threshold | 4.5100 | [3.9550, 4.5600] |
| | | 4.7400 | [4.5250, 4.7900] |

coefficient of the digital economy is 0.624 when the innovation environment threshold value is higher than 0.0662, which is significant at the level of 1%.

As shown in Table 9, when the institutional environment is lower than 4.510, the coefficient of the digital economy is 0.262, significantly positive at the 5% level. When the institutional environment is between 4.510 and 4.740, the coefficient of the digital economy is 1.762, significant at the 1% level; however, the coefficient of the digital economy is 0.618 when it is higher than 4.740, significant at the 1% level. The change of the digital economy coefficient increasing first and then decreasing indicates a significant threshold effect of the institutional environment. At the same time, it shows that the marginal returns of the digital economy, with the continuous improvement of the institutional environment, first increase and then decrease. In the process of the digital economy promoting the development of green total factor productivity, its marginal return is the largest when the institutional environment is between the critical value of 4.510 and 4.740, while marginal return of the digital economy on green total factor productivity is small in regions with low and high institutional environment.

## 4.6 Robustness check

To ensure the validity of the results, this study explores three robustness tests. First, since the level of digital economy in Beijing, Shanghai and Guangdong Province is significantly better

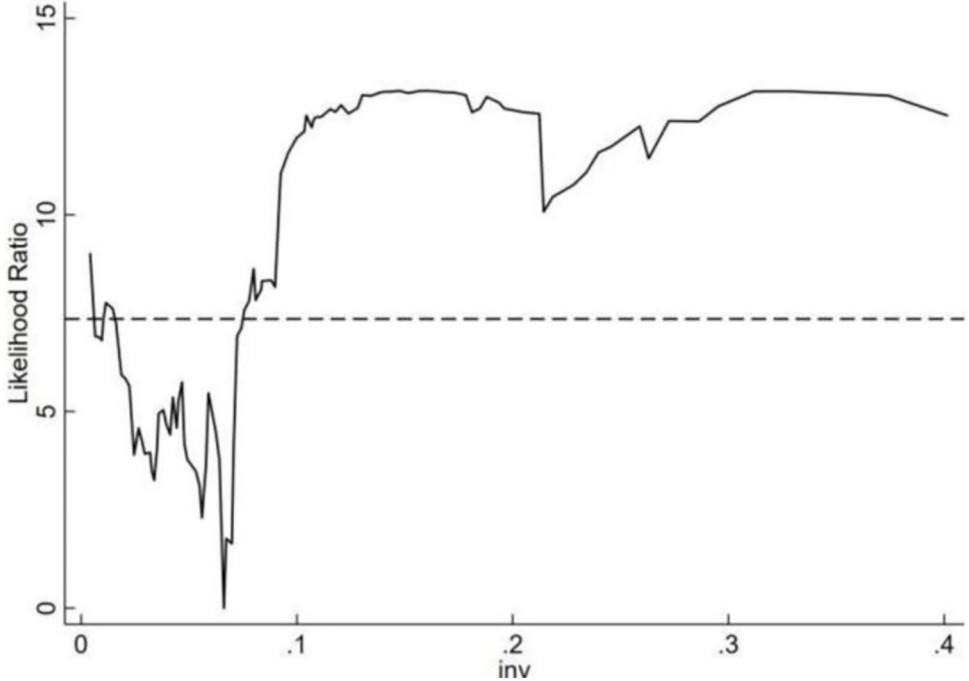

**Fig 1. Single threshold estimate and confidence interval in innovation environment model.**

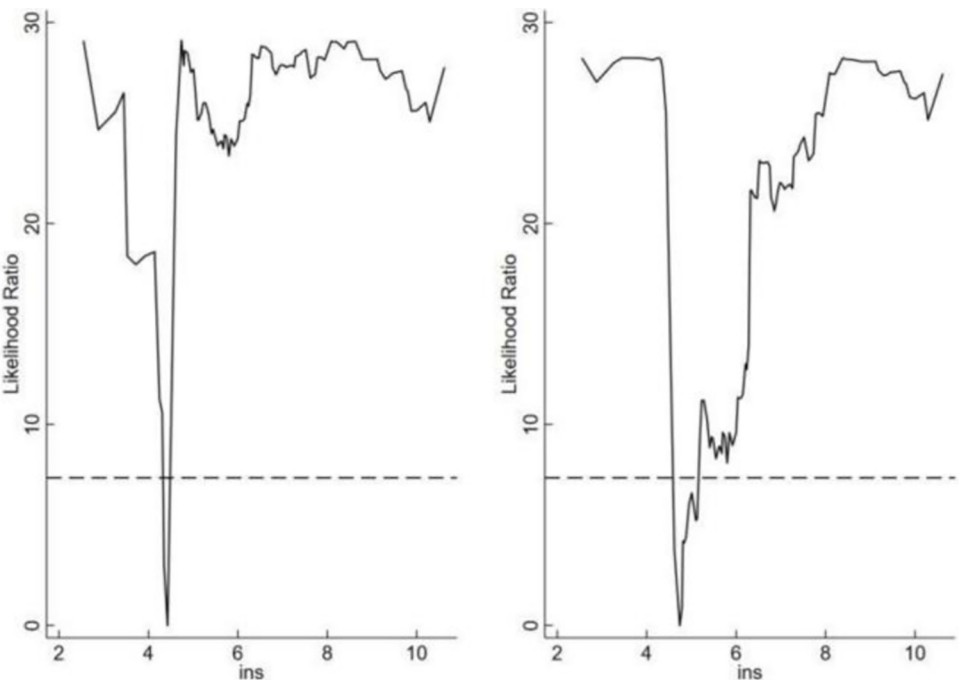

**Fig 2. Double threshold estimate and confidence interval in institutional environment model.**

than other provinces, this paper removes special samples and conducts empirical analysis. The estimated results are shown in column (15) of Table 10. Second, to ensure the reliability of the results and avoid chance in empirical results due to selection of specific models, we use the Reghdfe regression to retest the hypotheses in this paper. The results are shown in column (16) of Table 10. Third, the sample time is shortened to 2005–2019 for re-regression. The column (17) of Table 10 reports the results.

It can be seen from Table 10 that after excluding the samples of special provinces, the estimated coefficient between the digital economy and green total factor productivity is significantly positive. And the coefficient of green total factor productivity in column (15) and column (17) is significantly lagged by one period. At the same time, after replacing the regression model, neither the significance nor the direction of the coefficients have changed. After shortening the sample time, the results remain consistent. This paper confirms that provinces with a high level of digital economy have higher green total factor productivity. This robustness test is consistent with the expectation.

## 4.7 Further analysis

The quantile regression is used to analyze the change of the influence of the digital economy on green total factor productivity at different levels. The quantile regression is proposed by [50], which is a regression method to fit the linear function of the explanatory variable based on the conditional distribution of the explained variable. The quantile regression can not only avoid the interference of outliers, but also analyze the impact of the digital economy on different quantiles of green total factor productivity, and comprehensively describe the overall picture of the conditional distribution. This paper selects 4 representative quantiles of 20%, 40%, 60% and 80% to correspond to different green total factor productivity to analyze the impact

**Table 9. The result of threshold regression.**

| | (13) | | | (14) |
| --- | --- | --- | --- | --- |
| | **Innovation environment** | | | **Institutional environment** |
| road | 0.0040*** | | road | 0.0035*** |
| | (0.0090) | | | (0.0006) |
| edu | -0.0050 | | edu | -0.0108 |
| | (0.0030) | | | (0.0065) |
| rd | 0.0040 | | rd | 0.0018 |
| | (0.0030) | | | (0.0029) |
| indu | 0.0110 | | indu | -0.0025 |
| | (0.0150) | | | (0.0165) |
| pgdp | -0.0070 | | pgdp | 0.0075 |
| | (0.0080) | | | (0.0079) |
| open | 0.0740* | | open | 0.0580 |
| | (0.0411) | | | (0.0345) |
| for | 0.0099** | | for | 0.0112** |
| | (0.0042) | | | (0.0044) |
| tech | 0.0043 | | tech | 0.0054 |
| | (0.0055) | | | (0.0055) |
| inv≤0.0662 | -0.3130 | | ins≤4.510 | 0.2620** |
| | (0.2030) | | | (0.1020) |
| inv>0.0662 | 0.6240*** | | 4.5100<ins≤4.740 | 1.7620*** |
| | | | | (0.5820) |
| | (0.1550) | | ins>4.740 | 0.6180*** |
| | | | | (0.114) |
| cons | 0.9870*** | | cons | 0.9370*** |
| | (0.0860) | | | (0.0915) |
| N | 480 | | N | 480 |
| $R^2$ | 0.122 | | $R^2$ | 0.152 |

Note
*** $p<0.01$
** $p<0.05$
* $p<0.1$.

of the digital economy on green total factor productivity. The regression results are shown in Table 8. In the quantile regression, the regression coefficients of the digital economy are 0.1530, 0.2880, 0.3880, and 0.4650. From Table 11, it can be seen that the digital economy has a significant positive impact on green total factor productivity at different levels. It shows that when the green total factor productivity is different, the impact of the digital economy on green total factor productivity is significantly different. For provinces with high green total factor productivity, the digital economy can bring a more significant promotion effect. Therefore, the digital economy will widen the gap between green total factor productivity across provinces.

As shown in Fig 3, it is evident that the digital economy exerts a positive influence on the growth of green total factor productivity. The impact of the digital economy on the growth of green total factor productivity exhibits a consistent upward trend, signifying that as the level of digitization advances, the influence on the growth of green total factor productivity is on the rise.

**Table 10. Robustness check.**

| | (15) | (16) | (17) |
|---|---|---|---|
| L.gtfp | 0.1390 | | 0.1393** |
| | (0.0858) | | (0.0596) |
| de | 0.8231** | 0.472*** | 0.4077** |
| | (0.2316) | (0.087) | (0.1666) |
| road | 0.0054** | 0.003** | 0.0045** |
| | (0.0016) | (0.001) | (0.0017) |
| edu | -0.0205** | -0.005 | -0.0123 |
| | (0.0091) | (0.005) | (0.0101) |
| rd | 0.0006 | 0.001 | -0.0043 |
| | (0.0039) | (0.004) | (0.0050) |
| indu | -0.0128 | 0.013 | -0.0177 |
| | (0.0342) | (0.018) | (0.0310) |
| pgdp | -0.0092 | -0.001 | 0.0076 |
| | (0.0172) | (0.011) | (0.0156) |
| open | -0.0171 | 0.0487 | -0.0266** |
| | (0.0292) | (0.0391) | (0.0123) |
| for | 0.0069 | 0.0100* | 0.0046 |
| | (0.0056) | (0.0060) | (0.0058) |
| tech | 0.0136 | 0.0064 | 0.0068 |
| | (0.0096) | (0.0046) | (0.0106) |
| cons | 1.0524*** | 0.9826*** | 0.9209*** |
| | (0.1608) | (0.0832) | (0.1192) |
| N | 405 | 480 | 420 |
| $R^2$ | | 0.133 | |
| AR(1) | 0.002 | | 0.001 |
| AR(2) | 0.552 | | 0.493 |
| Hansen test | 0.633 | | 0.335 |

Note

*** $p<0.01$

** $p<0.05$

* $p<0.1$. Hansen statistic to test whether the moment condition has an over-identification problem to determine whether the instrumental variable is valid. AR(1) is the residual first-order serial correlation test. AR(2) is the residual second-order serial correlation test.

## 5. Discussion

As shown in Table 6, the digital economy can significantly enhance green total factor productivity, which is consistent with the results in the literature [51, 52]. This paper used Generalized Method of Moments method to prove that digital economy can promote the green total factor productivity. Not only does the digital economy boost economic efficiency, but it also induces a positive impact on environmental protection [53]. The digital economy plays an essential role in the enhancement of green total factor productivity for two primary reasons. First, the development of the digital economy serves as a drive for technological innovation and progress, such as big data, internet of things, and artificial intelligence are applied to traditional industries in various provinces and stimulate green output growth [54]. Second, as a new type of production factor, the digital economy can integrate with input factors such as capital and energy in green total factor productivity, lower energy consumption, enhance energy

**Table 11. The quantile regression.**

|  | (18) | (19) | (20) | (21) |
|---|---|---|---|---|
|  | Q20 | Q40 | Q60 | Q80 |
| de | 0.1530** | 0.2880** | 0.3880*** | 0.4650*** |
|  | (0.0764) | (0.1150) | (0.1490) | (0.1750) |
| road | -0.0002 | 0.0022** | 0.0050*** | 0.0062*** |
|  | (0.0011) | (0.0011) | (0.0011) | (0.0012) |
| edu | 0.0033 | 0.0020 | -0.0060 | -0.0213*** |
|  | (0.0080) | (0.0066) | (0.0072) | (0.0068) |
| rd | 0.0095* | -0.0001 | -0.0059 | -0.0133** |
|  | (0.0049) | (0.0050) | (0.0059) | (0.0054) |
| indu | 0.0034 | 0.0007 | 0.0180 | -0.0067 |
|  | (0.0194) | (0.0224) | (0.0221) | (0.0261) |
| pgdp | -0.0207* | -0.0019 | 0.0101 | 0.0251* |
|  | (0.0119) | (0.0113) | (0.0136) | (0.0147) |
| open | 0.0046 | -0.0107 | -0.0175 | -0.0245** |
|  | (0.0080) | (0.0100) | (0.0113) | (0.0120) |
| for | 0.0019 | 0.0036 | 0.0121** | 0.0153** |
|  | (0.0057) | (0.0046) | (0.0059) | (0.0078) |
| tech | 0.0001 | -0.0011 | 0.0020 | 0.0100* |
|  | (0.0034) | (0.0039) | (0.0038) | (0.0055) |
| Cons | 1.0460*** | 0.9980*** | 1.0090*** | 1.1040*** |
|  | (0.0662) | (0.0675) | (0.0779) | (0.0885) |
| N | 480 | 480 | 480 | 480 |

Note
*** p<0.01
** p<0.05
* p<a0.1.

utilization efficiency, transform the traditional extensive development mode, and explore new paths for green growth [55]. We found that the digital economy holds significant value in terms of green economics, which not only aligns with the principles of sustainable development but also boosts high-quality growth.

The impact of the digital economy on green total factor productivity exhibits a singular threshold effect in the innovation environment, which is consistent with the findings from previous studies. [56, 57] pointed that the development of the digital economy and the enhancement of green total factor productivity requires support from an innovation environment. In a sluggish innovation environment, provinces may be exposed to uncertainty in the process of economic growth. Inadequate impetus for digital technology gives rise to a tendency to allocate more important resources to high-profit production in order to reduce uncertainty in the external innovation environment. As a result, regions may not take advantage of the digital economy to improve green total factor productivity, and the digital economy exerts a significantly negative impact on green total factor productivity in this regard. Once the innovation environment exceeds a certain standard, digital economy advanced regions are more likely to invest in digital infrastructure and technological transformation [58]. With increased digitization, provinces will make use of the technological and innovative advantages of their own digital economy to phase out outdated production capacity, reduce pollution emissions, and enhance capacity utilization efficiency to boost the growth of green total factor productivity

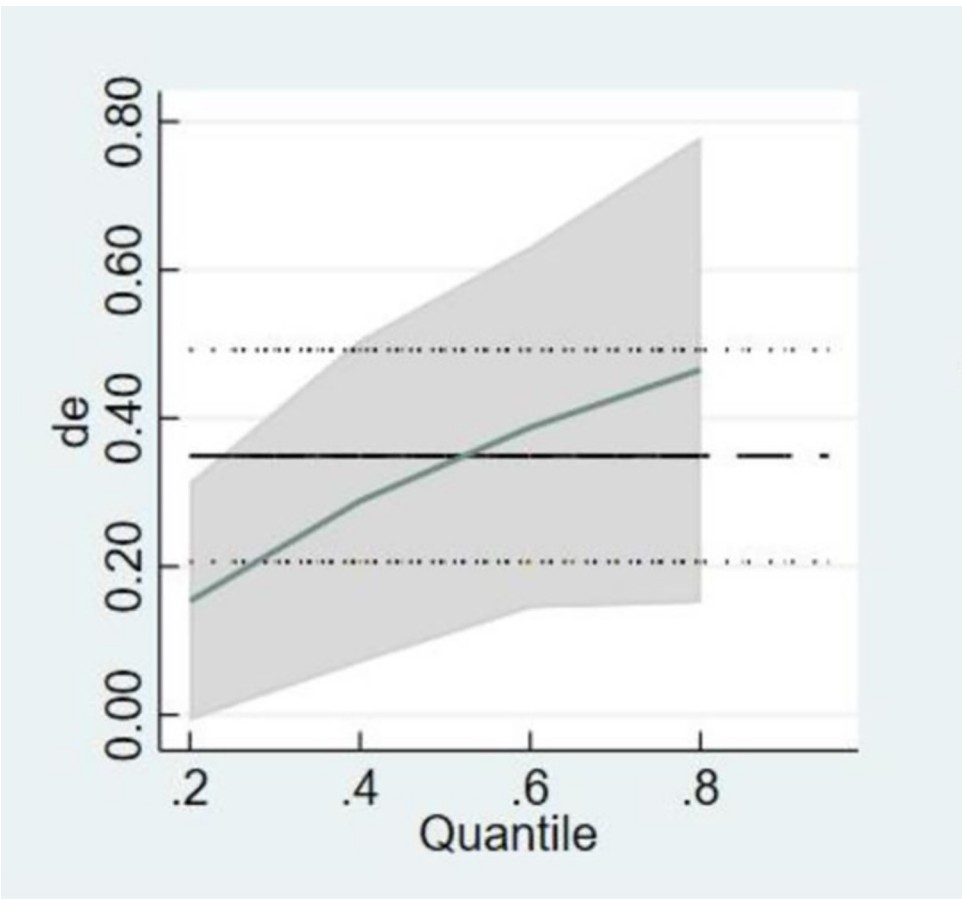

**Fig 3. The quantile regression graph of the influence of digital economy on green total factor productivity.** Note: In the figure, the horizontal axis represents different quantile points of the digital economy's impact on green total factor productivity, while the vertical axis represents the regression coefficients of the digital economy. The dashed line segments depict the OLS regression estimates of the digital economy, and the region between the two dotted lines represents the confidence interval of the OLS regression values (with a confidence level of 0.95). The solid line represents the quantile regression estimates of the digital economy, with the shaded area indicating the confidence interval of the quantile regression estimates (at a confidence level of 0.95).

[59]. As shown in Table 7, the innovation environment has a nonlinear effect on the digital economy and the improvement of green total factor productivity, which indicates a threshold effect.

As suggested by [60, 61], there is a positive and linear relationship between the institutional environment and the promotion of the digital economy and green total factor productivity. A well-functioning institutional environment implies a fair market environment and the rational distribution of green resources, which can help enhance green total factor productivity [62]. However, and contrary to our expectation, we found that there is significant difference in the digital economy and green total factor productivity between regions with a low degree, a medium and a high degree of institutional environment. The impact of the digital economy on green total factor productivity exhibits non-linear characteristics. Besides, in regions with medium to low degree of the institutional environment, the improvement of the institutional environment not only contributes to the full play of institutional advantages but also stimulates the vitality of market economic entities in various regions, increases investment in digital technology research and development, drive digital reform, optimize internal resource allocation,

and exert a positive influence on green total factor productivity of the digital economy, which demonstrates an increasing marginal returns of the digital economy. The marginal return of the digital economy in regions with a high level of institutional environment is lower than in regions with a medium-level institutional environment for the reason that the former boasts a more sophisticated legal foundation and market environment for the operation of the digital economy.

Given the relative scarcity of resources, they exhibit less tendency to allocate a large amount of vital resources to the development of the digital economy, which causes a diminishing marginal return as the level of digital economy development reaches a critical scale. In summary, the impact of the digital economy on green total factor productivity displays a dual threshold effect in the institutional environment.

## 6. Conclusions and research recommendations

The digital economy is an important factor in promoting the efficiency of the green economy. The digital economy plays an important role in promoting green total factor productivity. The panel data of 30 provinces in China are used in this study during the period of 2004–2019. This paper analyzes the impact of the digital economy level on green total factor productivity by using the system Generalized Method of Moments and the panel threshold model. The paper draws the following conclusions and the policy implications.

First, the digital economy has a significant positive impact on the improvement of green total factor productivity, and has become a new driving force for green economic growth. This conclusion is still true after excluding special provincial samples, replacing models, and shortening sample time. Therefore, the digital economy should be vigorously developed in various provinces, basic industries of the digital economy such as big data and blockchain should be cultivated, accelerating the development of the digital economy industry, improving the construction of digital infrastructure, and digital industries should be integrated with traditional industries such as high energy consumption and high pollution. Moreover, digital technology should be applied to the field of ecological environmental protection to reduce pollutant emissions and improve overall green total factor productivity.

Second, the impact of the digital economy on China's green total factor productivity has a single threshold effect with the innovation environment as the threshold variable. The impact of the digital economy on green total factor productivity has changed from negative to positive. It shows that regions with a higher level of innovation environment are more motivated to carry out digital technology innovation and digital infrastructure construction, and gain the advantages of green innovation brought by the digital economy. Therefore, the investment of scientific and technological innovation in the field of environmental protection should be strengthened, intensifying green technology research and development, developing innovative digital economy industries vigorously, promoting the development of "industry-university-research". Furthermore, the integration of digital economy and talent strategy, foreign investment, technological upgrading, etc. should be strengthened, improving the innovation environment, integrating digital innovation into traditional industries, so as to improve green total factor productivity and realize green economic development.

Third, the impact of the digital economy on China's green total factor productivity has a double threshold effect with the institutional environment as the threshold variable. The effect of the digital economy on green total factor productivity varies with the level of the institutional environment. And the positive effect is most significant when the institutional environment is between a certain interval value. When this value is crossed, the impact of the digital economy on green total factor productivity is weakened. This shows that the digital economy

presents a nonlinear characteristic of China's green total factor productivity from marginal increase to marginal decrease. Therefore, it is suggested to strengthen the government's guiding role in the market. On the premise of better playing the role of the government, it is necessary to stimulate the vitality and dynamism of enterprises and the market, improve the institutional environment as well as governance, and formulate effective policies to improve the development of the digital economy. Additionally, the development of innovative digital industries should be greatly supported, actively cultivating investment entities in a diversified manner. Furthermore, it is essential to increase investment in the digital economy, so that it can fully integrate with all walks of life based on the high permeability of the digital economy, effectively promoting resource utilization and efficiency. At the same time, allocation efficiency should be improved and pollutant emissions should be reduced, improving the ecological environment, and promoting the digital economy continuously to promote green economic development [63].

Fourth, further analysis finds that the impact of the digital economy on China's green total factor productivity has different quantile effects. The regression conclusions are generally consistent with the previous empirical evidence. Therefore, the development of the digital economy should be rationally planned in various regions, strengthening the construction of digital soft and hard environments, improving digital infrastructure such as 5G and big data, and narrowing the gap in the development of the regional digital economy. Moreover, the development of the digital economy should be steadily promoted according to local conditions and scientific guidance, so as to fully release the dividends of the development of the digital economy, and contribute to green and high-quality economic development.

## 7. Limitations and further research

There are several limitations to this paper. First, its broader connotations were not fully considered for the construction of digital economic indicators due to the absence of a unified definition of the digital economy, such as industrial digitization and digital industrialization. Consequently, the measurement system used in this study is unable to reflect the level of digital economic development comprehensively and accurately. Besides, there are inherent weaknesses in the construction of the indicator system. For future research, it is recommended to explore digital economy at multiple levels and examine their impact on green total factor productivity. Second, there are many factors and mechanisms that can improve green total factor productivity, which requires further investigation. Future studies should be conducted to investigate the interaction between internal and external factors, which can improve the understanding of green total factor productivity. Third, although the dynamic effects of digital economy on green total factor productivity are identified in this paper, the spatial spillover effects associated with digital economy are still not addressed. Future research can focus on exploring how to improve the methods of modelling and testing.

## Supporting information

**S1 Data.**
(XLSX)

## Author Contributions

**Data curation:** Yueping Zheng.

**Methodology:** Hailan Yang.

**Supervision:** Hailan Yang.

**Writing – original draft:** Yueping Zheng.

**Writing – review & editing:** Shuo Wang.

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
