## [Decision Letter · Decision Letter 0]

10 Apr 2023

PONE-D-22-25836Digital Economy and Green Total Factor Productivity in ChinaPLOS ONE

Dear Dr. zheng,

Thank you for submitting your manuscript to PLOS ONE. After careful consideration, we feel that it has merit but does not fully meet PLOS ONE’s publication criteria as it currently stands. Therefore, we invite you to submit a revised version of the manuscript that addresses the points raised during the review process.

Revise the manuscript as per reviewer comments. 

We look forward to receiving your revised manuscript.

Kind regards,

Abdul Majeed

Academic Editor

PLOS ONE

Journal Requirements:

2. PLOS ONE does not copy edit accepted manuscripts (https://journals.plos.org/plosone/s/criteria-for-publication#loc-5). To that effect, please ensure that your submission is free of typos and grammatical errors.

Reviewers' comments:

Reviewer's Responses to Questions

**Comments to the Author**

1. Is the manuscript technically sound, and do the data support the conclusions?

Reviewer #1: Partly

2. Has the statistical analysis been performed appropriately and rigorously? 

Reviewer #1: N/A

3. Have the authors made all data underlying the findings in their manuscript fully available?

Reviewer #1: Yes

4. Is the manuscript presented in an intelligible fashion and written in standard English?

Reviewer #1: Yes

5. Review Comments to the Author

Reviewer #1: This paper studies the relationship between the Digital Economy and Green Total Factor Productivity to promote high-quality regional development. The context of the research is novel, and it is very interesting. Furthermore, it has great motivation for the development of society. However, there are some questions. See below.

1. Is there a basis for the measurement of the digital economy and how is the selection of indicators made? What are the similarities and differences with the indicator designs currently considered in other studies? How do the results of the digital economy evaluation reflect consistency with reality?

2. The process of data analysis is a bit tedious. With only six variables, is it necessary to use the VIF test to test for multicollinearity? In addition, GMM estimation was chosen because of the significance of the coefficients? Is the estimated model calculated by GMM method significant? Please analyze the reasons for choosing GMM estimation and explain the advantages of GMM compared to other methods.

3. Please explain the philosophy of the robustness test. Are the three robustness analyses here intended to illustrate the robustness of the method or the robustness of the results? I don't quite understand the meaning of including this test here for the full analysis.

4. In addition, the second part of this paper has garbled codes and the formula expressions have disappeared.

5. Please check the focus of the article and sort out the analysis process of the article.

6. PLOS authors have the option to publish the peer review history of their article (what does this mean?). If published, this will include your full peer review and any attached files.

Reviewer #1: No

---

## [Author Response · Author response to Decision Letter 0]

18 May 2023

Response to Reviewers

We sincerely thank the editor and the reviewer for their valuable suggestions that we have used to improve the quality of our manuscript.The reviewer comments are laid out below in italicized font and specific concerns have been numbered.Our response is given in normal font.In this revised version, changes to our manuscript are all highlighted within the document by using blue-colored text. Point by point responses to the nice editor and the nice reviewer are listed below this letter.

Review Comments to the Author

This paper studies the relationship between the Digital Economy and Green Total Factor Productivity to promote high-quality regional development. The context of the research is novel, and it is very interesting. Furthermore, it has great motivation for the development of society. 

Response

We thank you for your appreciation of the study and we have seriously considered your primary concerns with our manuscript.According to the reviewers’comments, we have made extensive modifications.And we have defined the research framework and supplemented extra variables to make our results convincing. 

1. Is there a basis for the measurement of the digital economy and how is the selection of indicators made? What are the similarities and differences with the indicator designs currently considered in other studies? How do the results of the digital economy evaluation reflect consistency with reality?

Response

We appreciate the reviewer’s insightful comments.The reviewers' comments indicate that our exposition needs further fleshing out.We adopt the professional comments of the reviewer and provide the following explanation.Based on the meaning of digital economy, the Internet is an important carrier for the development of digital economy.The construction of digital economy indicators at the inter-provincial level takes Internet informatization as the core of the measurement.For example, Liu(2020)published an article on constructing the digital economy index through informatization and digitization indicators.For the selection of digital economy indicators, in line with the literature (Chen, 2022; Liu et al., 2020), our indicators include both total telecommunication services per capita, the share of the number of people in information computer services, and other information technology indicators.Informationization indicators reflect the current degree of informationization in the region and reflect the basic support environment of the development of the digital economy.Our indicators also include the number of Internet broadband access users, the number of cell phone users and other Internet development indicators, which fully consider the infrastructure of the digital economy and reflect the conditions for the development of the digital economy.

2. The process of data analysis is a bit tedious. With only six variables, is it necessary to use the VIF test to test for multicollinearity? In addition, GMM estimation was chosen because of the significance of the coefficients? Is the estimated model calculated by GMM method significant? Please analyze the reasons for choosing GMM estimation and explain the advantages of GMM compared to other methods.

Response

We appreciate the reviewer’s professional comments.we have followed the reviewer’suggestion by adding three control variables in all the models of the revised manuscript on the basis of prior literature (Line 419-426 of the revised manuscript), as below:(1) Foreign direct investment (for), foreign investment in regional green innovation affects the improvement of green total factor productivity.(2) the level of science and technology (tech), the higher the level of science and technology, the stronger the green technology innovation capacity, thus affecting the green total factor productivity.(3) The level of opening up (open), the higher degree of openness between the regions of the faster flow of factors of production, conducive to enhance green total factor productivity.In addition to the three control variables mentioned above, we used sample exclusion, model replacement, and time reduction for robustness testing.The results in columns (15)-(17) of Table 10 in the revised manuscript indicate that the results of our study are robust after adding control variables.

Thank you to the reviewer for your professional comments.We also note the necessity for multicollinearity tests and the issue of significance of GMM method.With your prompting, we make the following responses and modifications: (1) Regarding the necessity of VIF test. We include the VIF test in the article to mitigate the effect of multicollinearity.We have added some explanations for the VIF test results in the revised manuscript.(2) On the model selection and significance of GMM.The results of the estimated models we calculated using the GMM method are all significant.We chose the GMM method estimation not because of the significance of the coefficients.We choose GMM estimation for the following reasons: first, green total factor productivity incorporates resource and environmental factors into the analysis framework.Changes in resources and environmental factors are a dynamic process.The change in green total factor productivity in the previous period affects green total factor productivity in the current period.Therefore, to reflect the dynamic characteristics of green total factor productivity over time, we use the GMM method for estimation.Second, the explanatory variables contain the lagged terms of the explained variables.This can cause the explanatory variables to be correlated with the stochastic perturbation terms, which can lead to endogeneity problems.In this case, we use methods such as ordinary least square and maximum likelihood estimate leading to biased estimation results.The GMM approach overcomes the endogeneity problem in the regression process by using appropriate instrumental variables.Therefore, we use the GMM method.

we have followed the reviewer’s suggestion by adding vif test of the revised manuscript on the basis of prior manuscript (Line 457-465 of the revised manuscript), as below: “VIF tests of the main variables are provided in Table 4, intending to preliminarily check for the presence of multicollinearity. To verify there is no multicollinearity, the VIF test was conducted in this study. All variables have VIF < 5, and the mean VIF is 1.980, indicating that multicollinearity is not a problem. ” 

3. Please explain the philosophy of the robustness test. Are the three robustness analyses here intended to illustrate the robustness of the method or the robustness of the results? I don't quite understand the meaning of including this test here for the full analysis.

Response

The reviewer' comments give us important insights.The purpose of the robustness test in this paper needs further explanation.Robustness tests verify the reliability of the findings through a series of methods.Three robustness analyses are performed to illustrate the robustness of the results.Consistent with the literature (Han, 2021; LYu et al., 2022), in the first robustness analysis, we take the exclusion of special samples to verify the robustness of the results because Beijing, Shanghai and Guangdong have higher levels of digital economy than other provinces. In the second robustness analysis, to avoid significant results due to the choice of a specific model, we take a substitution model to verify the robustness of the results.In the third robustness analysis, we take a shortened time to verify the robustness of the results in order to avoid chance results due to the selection of a specific sample period.

4. In addition, the second part of this paper has garbled codes and the formula expressions have disappeared.

Response

We are very grateful to the reviewer for careful review.We have re-reviewed the submitted manuscripts.We found that the problem you pointed out due to the software version (file conversion).We apologize to you for this oversight.We have also done several checks elsewhere in the text to ensure that such problems do not occur again.

5. Please check the focus of the article and sort out the analysis process of the article.

Response

We thank the reviewer for your professional comments.The logical framework of the original manuscript need to be further clarified and the focus of the study sharpened.Based on the reviewer' comments, we have revised the introduction of the article (lines 41-128 of the revised version).We also adjusted the logical framework of the article.The details are as follows: The theoretical introduction is stated in Section 1.The introduction focuses on the research background of the digital economy and green total factor productivity.Based on this, we clarify the research methodology and address the relevant questions.Section 2 presents the literature review and research hypotheses.We combed through the literature on the digital economy and green total factor productivity to discover the innovative points of the study, and then proposed the research hypothesis.Section 3 describes the data sources and model construction.We construct indicators of the digital economy and green total factor productivity based on the data.We select the variables and thus construct dynamic panel models and threshold models.Section 4 analyzes the empirical results.We describe the impact of the digital economy on green total factor productivity and explore the role of the digital economy on green total factor productivity in terms of the innovation environment and institutional environment.We perform robustness tests to ensure the robustness of the results.Section 5 provides a discussion of further analysis. We use quantile regression to distinguish whether there is a difference in the impact of the digital economy on green total factor productivity when it varies.Finally, the conclusions and proposals for further studies stand in Section 6.We summarize the results of the previous study on the digital economy and green total factor productivity.Then, we make the corresponding recommendations.

Again, we thank you for all the above constructive comments.In addition, during the revision process, we added the necessary theoretical literature.We also checked the full text of words and phrases to further improve the standardization and readability of the paper.If you find any problems in the follow-up review, please let us know in time.We will certainly continue to revise and improve the paper according to your requirements in order to make it meet the publication standards of the Plos One journal.

---

## [Decision Letter · Decision Letter 1]

20 Sep 2023

PONE-D-22-25836R1Digital economy and green total factor productivity in ChinaPLOS ONE

Dear Dr. zheng,

Thank you for submitting your manuscript to PLOS ONE. After careful consideration, we feel that it has merit but does not fully meet PLOS ONE’s publication criteria as it currently stands. Therefore, we invite you to submit a revised version of the manuscript that addresses the points raised during the review process.

We look forward to receiving your revised manuscript.

Kind regards,

Abdul Majeed

Academic Editor

PLOS ONE

Reviewers' comments:

Reviewer's Responses to Questions

**Comments to the Author**

1. If the authors have adequately addressed your comments raised in a previous round of review and you feel that this manuscript is now acceptable for publication, you may indicate that here to bypass the “Comments to the Author” section, enter your conflict of interest statement in the “Confidential to Editor” section, and submit your "Accept" recommendation.

Reviewer #1: All comments have been addressed

Reviewer #2: (No Response)

2. Is the manuscript technically sound, and do the data support the conclusions?

Reviewer #1: Partly

Reviewer #2: No

3. Has the statistical analysis been performed appropriately and rigorously? 

Reviewer #1: Yes

Reviewer #2: No

4. Have the authors made all data underlying the findings in their manuscript fully available?

Reviewer #1: No

Reviewer #2: Yes

5. Is the manuscript presented in an intelligible fashion and written in standard English?

Reviewer #1: Yes

Reviewer #2: No

6. Review Comments to the Author

Reviewer #1: The overall content has been updated and improved. After carefully checking the related writing details and formulas, it can be accepted. Additionally, concerning issues related to publication, research ethics, or publication ethics, the author should pay attention. If there are similar studies related to these topics, please include them in the list of references.

Reviewer #2: Digital economy and green total factor productivity in China

Abstract:

1. Digital economy plays an important role in improving regional green total factor productivity and promoting high-quality economic development.

Do not claim the outcome at the start of the sentence.

2. Why this study is important: mention the gap before the objective.

3. Do not use abbreviations in the abstract like the SBM-DDF model.

4. According to a further research

Which future research can authors explain?

5. What is this institutional environment?

6. What’s the innovation environment?

Introduction:

7. Examples of such mechanisms include the relationship between the digital economy and technological innovation, financial infrastructure, environmental regulations, foreign direct investment, and more.

It is important to add the reference for the above statement.

8. Thus, this thesis aims to investigate the impact of the digital economy on green total factor productivity.

The thesis word is not appropriate here.

9. What is the DEA-SBM model?

10. The introduction of the should have motivation, key objectives and contribution in a clear way.

Literature review and study hypotheses

11. The literature review is not synchronized. It looks like the summary of different studies. So, it should be improved by mentioning what has been done on this topic and explaining the literature gap that will be covered in this study.

12. Economic theories should support the hypothesis of the study.

Research design

13. The research design of the study requires further explanation.

Data source and sample selection

14. It is important to provide the data source link of each variable in the study.

15. The green total factor productivity variable methodology has so many details. The authors explain how they combine the capital, energy, and labor inputs.

16. What’s the logic and reasoning for selecting the mentioned variables for the digital economy?

17. The Fan Gang market index measures the institutional environment, so it is important to give the details and link to this index.

18. How authors choose the control variables. What is the logic behind this? Are these the only variables that affect the TFP?

Empirical analysis

19. What’s the motivation to choose the fixed effect and GMM method?

20. Which IVs are used while applying the GMM method, and how are they suitable?

21. The results and discussion are weak. It is important to add the economic justification of the outcome and compare them with previous studies.

22. The quantile regression graph should be added for clear understanding.

Conclusions and research recommendations

23. The authors fail to conclude the study. The policy implication should be based on the findings. The limitations and future research directions are missing.

24. The language quality of the manuscript should be improved.

25. There should be symmetry in the whole manuscript.

7. PLOS authors have the option to publish the peer review history of their article (what does this mean?). If published, this will include your full peer review and any attached files.

Reviewer #1: No

Reviewer #2: No

---

## [Decision Letter · Decision Letter 2]

8 Dec 2023

PONE-D-22-25836R2Digital economy and green total factor productivity in ChinaPLOS ONE

Dear Dr. zheng,

Thank you for submitting your manuscript to PLOS ONE. After careful consideration, we feel that it has merit but does not fully meet PLOS ONE’s publication criteria as it currently stands. Therefore, we invite you to submit a revised version of the manuscript that addresses the points raised during the review process.

We look forward to receiving your revised manuscript.

Kind regards,

Dr. Jiachao Peng

Academic Editor

PLOS ONE

Additional Editor Comments:

The author should pay attention to the second round of revision comments and submit the responses incorporating these comments.

If the response to the second round of review comments is not submitted, the paper will be rejected.

The system does not have a record of a response to the second round of review comments.

Also, this paper lacks a discussion section, and the current content belongs to robustness testing.

Reviewers' comments:

Reviewer's Responses to Questions

**Comments to the Author**

1. If the authors have adequately addressed your comments raised in a previous round of review and you feel that this manuscript is now acceptable for publication, you may indicate that here to bypass the “Comments to the Author” section, enter your conflict of interest statement in the “Confidential to Editor” section, and submit your "Accept" recommendation.

Reviewer #1: All comments have been addressed

2. Is the manuscript technically sound, and do the data support the conclusions?

Reviewer #1: Yes

3. Has the statistical analysis been performed appropriately and rigorously? 

Reviewer #1: Yes

4. Have the authors made all data underlying the findings in their manuscript fully available?

Reviewer #1: Yes

5. Is the manuscript presented in an intelligible fashion and written in standard English?

Reviewer #1: Yes

6. Review Comments to the Author

Reviewer #1: 1. The article appears to be somewhat lengthy, please condense it and make the targrt of this article more clear. This includes the introduction, hypotheses development, data source, and sample selection. For the latter, consider presenting the variables in a list or using a more concise and clear format.

2. Please review the overall grammar in the article. Ensure universal in the formatting of the entire text and the references, and space before and after text, punctuation.

7. PLOS authors have the option to publish the peer review history of their article (what does this mean?). If published, this will include your full peer review and any attached files.

Reviewer #1: No

---

## [Author Response · Author response to Decision Letter 2]

13 Jan 2024

Responses to Reviewer

According to the system flow, we pay attention to the second round of revision comments and we submit responses incorporating these comments in Responses to Reviewers. In addition, we add a discussion section to the paper. The detailed discussion section is shown in Revised Manuscript with Track Changes.

We sincerely thank the editor and the reviewer for your valuable suggestions that we have used to improve the quality of our Manuscript. The reviewer’s comments are laid out below in italicized font and specific concerns have been numbered. Our response is given in normal font. In this revised version, point by point responses to the editor and the reviewer are listed below this letter.

Reviewer’s Comments to the Author

1. The article appears to be somewhat lengthy, please condense it and make the targrt of this article more clear. This includes the introduction, hypotheses development, data source, and sample selection. For the latter, consider presenting the variables in a list or using a more concise and clear format.

Response

We feel great thanks for your profeessional review work on our paper. As the suggested by the reviewer, we have made the following modifications. We simplify the introduction and the hypotheses development (Line 39-87 and Line 140-219 of the Revised Manuscript with Track Changes). We use tables to present the variables (Line 245-273 of the Revised Manuscript with Track Changes).

2. Ensure universal in the formatting of the entire text and the references, and space before and after text, punctuation.

Response

We sincerely thank you for your careful comments. We are really sorry for our careless mistakes. As the suggested by the reviewer, we have made the corrections to make the formatting of the entire text and the references harmonized within the whole Manuscript. We have also checked the space before and after text, punctuation and made changes.

 Responses to the second round of review comments

We sincerely thank the editor and the reviewers for their valuable suggestions that we have used to improve the quality of our Manuscript. The reviewers’ comments are laid out below in italicized font and specific concerns have been numbered. Our response is given in normal font. In this revised version, changes for reviewer #2’s comments to our Manuscript are highlighted within the document by using blue-colored text. Point by point responses to the editor and the reviewers are listed below this letter. The detailed discussion section is shown in Manuscript.

Reviewer #1’s Comments to the Author

The overall content has been updated and improved. After carefully checking the related writing details and formulas, it can be accepted. Additionally, concerning issues related to publication, research ethics, or publication ethics, the author should pay attention. If there are similar studies related to these topics, please include them in the list of references.

Response

We thank you for your appreciation of the study and we have seriously considered your primary concerns with our Manuscript. According to the reviewer #1’s comments, we have made extensive modifications. We pay more attention to issues related to publication, research ethics, or publication ethics. And we have tried to find similar studies related to these topics. We have supplemented these studies to the list of references. The references are as follows:

Lyu Y, Wang W, Wu Y, Zhang J. How does digital economy affect green total factor productivity? Evidence from China. Science of The Total Environment. 2023; 857: 159428. 

Zhang C, Wang L, Yu S, Zhang Y. Energy production factors and total factor productivity in China: Evidence from a new perspective. Energy Economics. 2019; 78: 42–52.

Xiaoqi L, Enshuo L. Spatial Effects of digital economy and foreign direct investment—A u-shaped relationship. Information Systems and Economics. 2023; 4: 30–38.

Xu J, Li W. Study on the impact of digital economy on innovation output based on dynamic panel data model. European Journal of Innovation Management. 2023.

Reviewer #2’s Comments to the Author

1. Digital economy plays an important role in improving regional green total factor productivity and promoting high-quality economic development. Do not claim the outcome at the start of the sentence.

Response

We appreciate the reviewer#2’s insightful comments. We shouldn't claim the outcome at the start of the sentence. Therefore, we have followed the reviewer#2’s suggestion by revising the sentence of the Manuscript (Line 9-11 of the Manuscript), as below: “The development of information technology has created conducive conditions for the digital economy. The digital economy is regarded as a critical pathway for transforming traditional economic models.”

2. Why this study is important: mention the gap before the objective.

Response

We adopt the reviewers' suggestions, and add the importance of this study and the gap before the objective in the Manuscript (Line 11-15 of the Manuscript), as below: “Green total factor productivity serves as an indicator for assessing the quality of economic development. During pivotal periods of economic transition, the digital economy and green total factor productivity have emerged as two prominent themes for achieving sustainable economic development. But the impact of digital economy on green total factor productivity is less discussed.”

3. Do not use abbreviations in the abstract like the SBM-DDF model.

Response

We are very grateful to the reviewer#2 for careful review. We have re-reviewed the submitted manuscript. We found that the problem you pointed out due to our carelessness in the format of writing. We apologize to you for this oversight. We have also done several checks elsewhere in the text to ensure that such problems do not occur again. The full name of SBM-DDF model is the Slack based measure-Directional distance function model. (Line 21-22 of the Manuscript)

4. According to a further research Which future research can authors explain?

Response

We have followed the reviewer’s suggestion by adding a lot of explanations about the future research in the Manuscript (Line 23-29 of the Manuscript), as below: “Generalized Method of Moments model is constructed to carry out an empirical study on the impact of digital economy on green total factor productivity. This paper constructs a panel threshold model with innovation environment and institutional environment as threshold variables. In further analysis, this paper employs panel quantile regression for the empirical analysis of the impact of the digital economy on green total factor productivity. Further analysis elucidates the evident disparities in the influence of the digital economy on green total factor productivity at various levels.”

5. What is this institutional environment?

Response

We appreciate the reviewer#2’s insightful comments. We have followed the suggestion. We explain the definition of institutional environment in the abstract. We add the sentence of the Manuscript (Line 15-16 of the Manuscript), as below: “The institutional environment encompasses the aggregate of economic, political, social, and legal rules.”

6. What’s the innovation environment?

Response

We appreciate the reviewer#2’s insightful comments. We have followed the suggestion. We explain the definition of innovation environment in the abstract. We add the sentence of the Manuscript (Line 17-18 of the Manuscript), as below: “Innovation environment refers to a confluence of conditions shaped by factors such as talent, funding, cultural atmosphere and government policies, all of which collectively support innovative activities within a region.”

Introduction

7. Examples of such mechanisms include the relationship between the digital economy and technological innovation, financial infrastructure, environmental regulations, foreign direct investment, and more. It is important to add the reference for the above statement.

Response

We have followed the reviewer#2’s suggestion by adding the references about the relationship between the digital economy and technological innovation, financial infrastructure, environmental regulations, foreign direct investment. The references are as follows:

Li J, Chen L, Chen Y, He J. Digital economy, technological innovation, and green economic efficiency—Empirical evidence from 277 cities in China. MDE Manage Decis Econ. 2022;43: 616–629. doi:10.1002/mde.3406

Veselovsky MY, Pogodina TV, Ilyukhina RV, Sigunova TA, Kuzovleva NF. Financial and economic mechanisms of promoting innovative activity in the context of the digital economy formation. J Entrep. Sustain Issu. 2018;5: 672–681. doi:10.9770/jesi.2018.5.3(19)

Wang J, Zhang G. Can environmental regulation improve high-quality economic development in China? The mediating effects of digital economy. Sustainability. 2022;14: 12143. doi:10.3390/su141912143

Spatial effects of digital economy and foreign direct investment—A U-shaped relationship. infse. 2023;4. doi:10.23977/infse.2023.040205

8. Thus, this thesis aims to investigate the impact of the digital economy on green total factor productivity. The thesis word is not appropriate here.

Response

We appreciate the reviewer#2’s careful comments. We have followed the reviewer’s suggestion by replacing the word “thesis” with the word “paper”. Therefore, the revised sentence is “Thus, this paper aims to investigate the impact of the digital economy on green total factor productivity.” (Line 142-145 of the Manuscript)

9. What is the DEA-SBM model?

Response

We thank the reviewer for your professional comments. The explanation is as follows “The full name of DEA-SBM is data envelopment analysis - slack based measure. The DEA-SBM model is a method used in the field of operations research and production management to assess the efficiency of decision-making units. DEA is a non-parametric linear programming technique that evaluates the efficiency of DMUs by comparing them against a frontier formed by the best-performing units. The DEA-SBM model addresses this issue by incorporating these slacks directly into the efficiency score calculation. It provides a more comprehensive measure of efficiency by not only considering the radial distance but also the slacks in inputs and outputs.” (Line 84-90 of the Manuscript)

10. The introduction of the should have motivation, key objectives and contribution in a clear way.

Response

We are very grateful to the reviewer#2 for careful review. We have followed the reviewer#2’s suggestion by revising the introduction extensively. The content is as follows“Despite prior studies conducted to reveal the impact of digital economy on green total factor productivity, there remains room for improvement in terms of indicator measurement and theoretical frameworks. Firstly, environmental pollution and resource consumption have not been included as part of the total factor productivity indicator, although China's total factor productivity have been measured in some studies [6, 7]. Therefore, traditional indicators have been modified and upgraded to measure China's green total factor productivity more accurately. Secondly, the "clean" attribute of digital economy plays a crucial role in driving the growth of green economy. However, the focus of previous research is placed on the improvement of economic efficiency [8], with the environmental perspective of the digital economy ignored. Therefore, this study aims to find out the relationship between digital economy and green total factor productivity. To achieve this purpose, the panel data from 2004 to 2019 for 30 provinces in China is used to measure their green total factor productivity respectively. Besides, Generalized Method of Moments model and threshold model are applied to empirically examine the effect of digital economy on green total factor productivity, along with the threshold effects of innovation environment and institutional environment.

The potential theoretical contributions of this study are detailed as follows. First, the DEA-SBM model is used to include relevant undesirable outputs in traditional total factor productivity, and the Malmquist-Luenberger productivity index is used to classify green total factor productivity by region. With new data perspectives provided, this contributes to the existing literature. Second, the existing literature on the green effects of digital economy is expanded from a macro perspective by exploring the impact of the digital economy on green total factor productivity. Finally, innovation environment and institutional environment are introduced as threshold variables to explore their influence exerted on the relationship between the digital economy and green total factor productivity. This is beneficial in deepening our understanding as to the green economic consequences of digital economy from the perspectives of institutional environment and innovation environment. ”

Literature review and study hypotheses

11. The literature review is not synchronized. It looks like the summary of different studies. So, it should be improved by mentioning what has been done on this topic and explaining the literature gap that will be covered in this study.

Response

We are very grateful to the reviewer#2 for careful review. We have followed the reviewer#2’s suggestion by revising literature review extensively. The content is as follows“ As an emerging economy, the digital economy is having a significant impact on the way the economy is developed today. The digital economy comprises a series of economic activities that rely on digital knowledge and information as key production factors, with modern information networks serving as important carriers. Effective utilization of information and communication technology plays a critical role in driving efficiency improvements and optimizing economic structure. In contrast to traditional economies, the digital economy pervades every aspect of social production. While existing research on the digital economy has primarily centered on its economic consequences, more recent efforts have focused more directly on two critical aspects of digital economy research. The first critical aspect centers on a shift away from direct analysis of the digital economy's economic consequences to a more focused study of intermediate mechanisms. Examples of such mechanisms include the relationship between the digital economy and technological innovation, f

---

## [Decision Letter · Decision Letter 3]

15 Feb 2024

Digital economy and green total factor productivity in China

PONE-D-22-25836R3

Dear Dr. zheng,

We’re pleased to inform you that your manuscript has been judged scientifically suitable for publication and will be formally accepted for publication once it meets all outstanding technical requirements.

Kind regards,

Dr. Jiachao Peng

Academic Editor

PLOS ONE

Additional Editor Comments (optional):

Reviewers' comments:

Reviewer's Responses to Questions

**Comments to the Author**

1. If the authors have adequately addressed your comments raised in a previous round of review and you feel that this manuscript is now acceptable for publication, you may indicate that here to bypass the “Comments to the Author” section, enter your conflict of interest statement in the “Confidential to Editor” section, and submit your "Accept" recommendation.

Reviewer #1: All comments have been addressed

2. Is the manuscript technically sound, and do the data support the conclusions?

Reviewer #1: Yes

3. Has the statistical analysis been performed appropriately and rigorously? 

Reviewer #1: Yes

4. Have the authors made all data underlying the findings in their manuscript fully available?

Reviewer #1: Yes

5. Is the manuscript presented in an intelligible fashion and written in standard English?

Reviewer #1: Yes

6. Review Comments to the Author

Reviewer #1: After this last revision, the article has been perfected enough. The paper is recommended for acceptance in PLOS ONE.

7. PLOS authors have the option to publish the peer review history of their article (what does this mean?). If published, this will include your full peer review and any attached files.

Reviewer #1: No

---

## [Editor Report · Acceptance letter]

21 Feb 2024

PONE-D-22-25836R3 

PLOS ONE

Dear Dr. Zheng, 

I'm pleased to inform you that your manuscript has been deemed suitable for publication in PLOS ONE. Congratulations! Your manuscript is now being handed over to our production team.

Kind regards, 

on behalf of

Dr. Jiachao Peng 

Academic Editor

PLOS ONE